# Verbs in Mothers’ Input to Six-Month-Olds: Synchrony between Presentation, Meaning, and Actions Is Related to Later Verb Acquisition

**DOI:** 10.3390/brainsci7050052

**Published:** 2017-04-29

**Authors:** Iris Nomikou, Monique Koke, Katharina J. Rohlfing

**Affiliations:** 1Department of Psychology, Center of Situated Action and Communication, University of Portsmouth; King Henry Building, King Henry I Street, Portsmouth PO1 2DY, UK; 2Faculty of Arts and Humanities, Psycholinguistics, Paderborn University, Warburger Street 100, Paderborn 33098, Germany; mkoke2@campus.uni-paderborn.de (M.K.); katharina.rohlfing@uni-paderborn.de (K.J.R.)

**Keywords:** action–language synchrony, social interaction, verb acquisition

## Abstract

In embodied theories on language, it is widely accepted that experience in acting generates an expectation of this action when hearing the word for it. However, how this expectation emerges during language acquisition is still not well understood. Assuming that the intermodal presentation of information facilitates perception, prior research had suggested that early in infancy, mothers perform their actions in temporal synchrony with language. Further research revealed that this synchrony is a form of multimodal responsive behavior related to the child’s later language development. Expanding on these findings, this article explores the relationship between action–language synchrony and the acquisition of verbs. Using qualitative and quantitative methods, we analyzed the coordination of verbs and action in mothers’ input to six-month-old infants and related these maternal strategies to the infants’ later production of verbs. We found that the verbs used by mothers in these early interactions were tightly coordinated with the ongoing action and very frequently responsive to infant actions. It is concluded that use of these multimodal strategies could significantly predict the number of spoken verbs in infants’ vocabulary at 24 months.

## 1. Introduction

Embodied theories on language assume a direct link between language and action. For example, according to action-based language theory [1], experience in acting generates an expectation of this action when hearing the word for it. A translation of the embodied theory approaches to ontogenetic development implies that children need to have had a specific sensorimotor experience of a specific action before they can acquire a word for it (e.g., [2,3]). In their review, Wellsby et al. [3] (p. 7) suggest that what is relevant as a “sensorimotor experience” might depend on the word class that is learned. Another important issue is the source of this sensorimotor experience. This is the focus of the present article. Existing research has mostly focused on one source that provides children with this crucial experience: their own exploration. However, this view excludes another source which is omnipresent throughout children’s development: social interaction. More specifically, the suggestion put forward in this article is that social interaction is a tuning device for the infants’ sensorimotor experiences. This tuning is a gradual collaborative experience of repeated interactions with caregivers and other people. Through interaction the child learns to notice those aspects of his or her experience in the world which are relevant for the language to be learned. In the following, we shall sketch research on both infant exploration and social interaction. However, we shall elaborate on social interaction as a source of sensorimotor experience, because this perspective is still underrepresented.

One possible way for children to experience action is through their own exploration of the physical environment. In fact, approaches on cognitive development assume that children build up their experience gradually by discovering different aspects of events. Along these lines, Mandler [4] has emphasized that actions are central to the development of conceptualization and that they provide a solid basis for the acquisition of verbs. However, it has been suggested that translating this experience into verbal behavior is especially challenging, because it requires infants to have the ability to extract commonalities from dynamic action events and map particular aspects of actions onto the words offered to them in a target language [2]. This conception of acquisition—namely a mapping of a word onto a concept of action—contributes to the belief that verbs are acquired later than nouns, because categorizing dynamic movements across examples requires more abstraction for the former than the latter. Recent studies investigating children’s early understanding of words consider some verbs to be rather abstract words, and claim that in the presence of a referent, they occur less in the input (e.g., [5]). An investigation of the naturalistic input to young children might cast light on this issue and reveal whether the referents of verbs provided to children are of abstract or concrete nature.

Whereas the belief that verbs are acquired earlier than nouns has been proposed by [6] and Ambridge et al. [7] cast reasonable doubts on the empirical basis for this. In fact, the SICI model proposed by Golinkoff and Hirsh-Pasek [8] suggests that verbs are acquired concurrently with nouns. SICI stands for four factors that contribute to the learning of words: Shape, Individuation, Concreteness, and Imageability. The authors [8] proposed that the more concrete an entity is, the earlier the acquisition of a word for it, regardless of whether it is a noun or a verb. Hence, in contrast to the body of research suggesting that verbs are acquired later than nouns, SICI offers an alternative proposing that some verbs might be acquired earlier than some nouns. This may be due to the structure of a specific language. For example, in her work on the Tzeltal language, Brown [9] speaks of “heavy verbs”—with contextually rich semantics such as object incorporation—that might be easier to acquire than semantically sparse (or more abstract) verbs. Because verbs may incorporate properties of their objects in Tzeltal, acquiring their meaning does not require the same degree of abstraction from the situation as that needed with more general verbs. In this case, individuation or concreteness might be provided by the language structure itself. 

Another way of experiencing actions as a foundation for verb meaning is within social interaction. In this article, we focus on this aspect. Already in very early interactions, caregivers build their interactional behavior on infants’ predispositions [10]. This enables a reciprocal interaction early in a child’s development [11]. Recently, Nomikou and colleagues [12] have proposed that this reciprocity is achieved not only by caregivers stimulating their infants but also by infants responding to the caregivers’ behaviors—an interactive view suggested by observing mutual regulation and alignment in an interaction (e.g., [13]). 

Furthermore, studies have shown that caregivers act in a way that fits the infants’ own mechanisms of perceptual development [14]. One suggestion for the mechanism driving infants’ perception is temporal synchrony. This refers to the presentation of temporally overlapping intersensory information [15,16]. According to According to Bahrick and colleagues [17,18], early in development, infants selectively attend to intermodally specified events that are presented redundantly, because this way of presenting enhances their perceptual processing, learning, and eventually memory for multimodally specified properties. Importantly for our argument, the value of intersensory facilitation can also be observed in social interaction when mothers talk to their children and move themselves or other objects in synchrony with their vocal behavior [18,19,20]. A simultaneous presentation of language and action involves an interplay of the modalities involved: On the one hand, it helps infants to create a link between what words mean and what they refer to in the world, thus educating their attention toward relevant aspects of speech [14]; on the other hand, it helps infants parse the world into meaningful events, thus educating their attention toward understanding actions—an idea known as acoustic packaging pushed forward by Hirsh-Pasek and Gollinkoff [21] as well as Hollich and colleagues [22]. Investigating how language packages action, Brand and Tapscott [23] demonstrated the positive effects of the language–action link on action understanding by showing that 9.5-month-old infants discern different movements (e.g., A and B) as forming a sequence (AB) when both occur within one verbal narration. The assumption that language–action links facilitate the word–referent link has been reported in several studies presenting, for example, a label together with a synchronous movement of the referent [19]. According to Gogate and Bahrick [20], infants at the age of seven months could remember the link better than in an asynchronous presentation condition. Along the same lines, in an eye-tracking study, Rader and Zukow-Goldring [24] revealed that when novel words were presented in synchrony with dynamic gestures and speech, infants aged 9–14 months learned them better.

Beyond experimental research, benefits have also been observed in seminaturalistic studies: Rohlfing and Nomikou [25] suggested that the way in which mothers synchronize their vocal behavior with nonverbal actions constitutes a form of multimodal responsivity when this synchrony is provided in a timely way so that it fits with the infant’s gaze toward the mother. The authors found that the synchrony between action and language as a form of mothers’ multimodal responsive behavior at six months related to the child’s productive vocabulary at 24 months [25]. Findings were especially pronounced for the acquisition of verbs. Tamis-LeMonda and colleagues [26] proposed that responsiveness acts on two levels: On a social-interactive level, responsive input creates a pragmatic frame in which language is constructed out of shared, attuned activities with caregivers; on a sensory level, responsive input is temporally contiguous and conceptually contingent to infant behavior, thereby creating rich time-locked experiences. A similar argument was raised by McGillion and colleagues [27] when reporting that input that is both semantically appropriate and temporally linked to infant behavior at 9.5 months was related to infant expressive vocabulary at 18 months. Therefore, the available research suggests that temporally contiguous responsive behavior, which originates from multiple modalities and is provided in a temporally coordinated manner, can scaffold verb learning because it matches the infant’s mechanisms of perceptual learning [18].

In this article, we want to further explore action–language synchrony in input as a basis for infants’ experience with actions and later language development. Therefore, we focused on mothers’ use of verbs in interactions with six-month-old infants and addressed three main issues: First, we wanted to explore the ways in which verbs were embedded in the ongoing interaction with the infants in order to determine the concrete or abstract nature of the entities to which they refer and whether there is a relation between verbs referring to the ‘here and now’ and their temporal tightness to the accompanying actions. Following the SICI model, we reasoned that the more the verbs are temporally aligned with the ongoing action, the more concrete their reference would be. Second, we were interested in whether mothers’ behavior is regulated by the behavior of the infant. Based on the findings reported by [25,28], we were interested in whether they would be sensitive to the gaze direction of their infant. Third, we were interested in the relationship between the way verbs are embedded in the ongoing action and the children’s later language development. Following up on our previous work [25], we assumed that the synchronous presentation of verbs and action needs to be enhanced by being coupled to infants’ gaze in order to function as the meaningful and rich input that could guide acquisition processes.

## 2. Materials and Methods

We examined these issues in a corpus of natural early mother–infant interactions (for a detailed description, see [29]). This study was carried out in accordance with the recommendations of the Ethics Committee of the University of Muenster, Germany (2010-589-f-S). In accordance with the Declaration of Helsinki, from all subjects, a written informed consent was obtained. 

Focusing on the ways in which the mothers’ use of verbs was embedded in the ongoing interaction, we started by collecting single cases and analyzing them qualitatively. This led to the development of categories with which we then coded the maternal behavior within the entire corpus. In the next step of our investigation, we related the behavioral findings to parental reports on the infants’ productive vocabulary at 24 months. Data were gathered with the ELFRA-2 [30], a parental questionnaire (German adaptation of the Infant Form of the MacArthur-Bates Communicative Development Inventories). With respect to infants’ reported productive vocabulary, we were interested in looking at the number of spoken verbs to see whether mothers’ use of verbs early in a child’s development could predict the child’s later production of verbs.

### 2.1. Setup

Data were collected in the families’ homes by recording the everyday routine activity of diaper changing. The aim was to collect ecologically valid data outside laboratory conditions. The activity for the interaction was meant to be both age appropriate and familiar to mother and infant, drawn from their everyday activities. For this reason, we chose the context of diaper changing. In addition, lying down on the changing table is a natural position, giving the infant the opportunity to control her or his own movement independently without needing to be supported by the mother. This is important, as the infant was free to direct his or her attention to whatever he or she wanted and was free to engage or disengage from interaction with the mother.

Although the activity itself was kept as natural as possible, a changing table was used to ensure that the constraints on the mothers’ body posture were comparable across dyads. This foldable table was made of wood and 90 cm high. Mothers stood in front of it when changing their baby’s diaper. Two HD video cameras (Canon HF10, Canon Inc.; Tokyo; Japan, Sony 3CMOS HDV 1080i; Sony Corporation; Tokyo; Japan) mounted on camera stands recorded the activity: One camera was positioned behind the changing table (opposite where the mother was standing), filming from the bottom and slightly offset from the center. This camera was directed toward the mother’s face and upper body. The second camera was set up in front of the table behind the mother, thus filming at a higher level over her shoulder. This second camera captured the mother’s arms and the infant’s body from the side together with the infant’s hands, body, face, and gaze (see Figure 1). An external microphone was mounted on one of the cameras to ensure high-quality audio recording. The second camera recorded sound via its built-in microphone. Recordings were made at times the mother considered appropriate. Typically, this was after the infant had slept and been fed. Although appointments were scheduled some weeks before, mothers were encouraged to call the experimenter on the day of the filming in case they wanted the session to be held earlier or later. Recording was carried out at 25 frames per second. The mean duration of recordings was 481 s, (*SD* = 287 s).

### 2.2. Participants

We were able to collect parental reports on 14 of the 17 infants participating in the original study when they were aged 24 months. Only these dyads were included in the following analysis. Out of nine boys and five girls, 10 were firstborn. All had been born with no complications and were of average healthy birthweight. The average infant age was six months and four days (*SD* = 7 days). All mothers spoke German to their infants. During the recording period, all but one of the mothers were on parental leave.

### 2.3. Analysis and Coding

Our data analysis commenced with a qualitative description of single cases in a search for similarities between them that could be used to develop measurements to account for all cases; that is, objective coding categories. This enabled us to uncover the different qualities of the multimodal ways in which the mothers were embedding verbs. These observations then served as a basis for describing the data quantitatively in the next step of our analysis.

#### 2.3.1. Qualitative Analysis

One striking observation was that the mothers spoke about their actions while performing them (see also [19]). In this case, the temporal coordination between the words and the actions was very tight. The meaning of the verb was conveyed by the action taking place. A more detailed look at the different types of such coordination between verb and action revealed the following patterns: (a) an action could be perfectly timed with the verb; different forms of this timing are illustrated in Examples 1–3; (b) the verb could be used just before the action thereby, announcing it (Examples 4 and 5); (c) the verb could be positioned directly after an action thereby commenting it (Examples 6–8); and, finally, (d) the relationship between verb and action could be such that there was no perceivable connection between the meaning of the verb and the activity taking place (Example 9). These observations will be elaborated below through detailed description of some example cases. 

##### Perfectly Timed Action

One pattern involved mothers describing their own action or even talking about an action carried out on the body of the infants while performing it. In this case, there was not only audio–visual synchrony but also synchronous tactile and proprioceptive sensory information as illustrated in Example 1.

**Example 1.** VP06_4T (00:05:23.919–00:05:25.766)

The mother is undressing the infant. She accompanies the lifting movement of the infant’s body with her utterance. The infant is experiencing herself being lifted while the mother is uttering the word (Figure 2).

Mother:*Alles bitte einmal anheben.*
*(Please lift up everything.)*


**Example 2.** VP08_4T (00:01:06.912–00:01:08.842)

The mother turns around and picks up a clean diaper. While picking up the diaper, she announces what she is doing.

Mother:So, dann nehmen wir mal die Pampers.(So then, we take the diaper.)

**Example 3.** VP13_4T (00:02:18.462–00:02:19.368)

The infant has grasped the cream and is holding it. While taking it, the mother asks the infant if he is going to give it back to her. Interestingly, in this example, the mother is naming the action from the perspective of the infant. She is not saying “I take” but “you give.”

Mother:Gibst du mir das wieder?(Will you give it back to me?)

##### Announcing Action

A further observed pattern of action–verb coordination involved the mother using a verb and subsequently carrying out the relevant action. In this case, the verb was not perfectly timed with the action but announced the action to be performed.

**Example 4.** VP09_4T (00:03:06.316–00:03:07.854)

In this example, the mother takes the infant’s slipper in her hand, and while holding it in front of the infant, she stops and announces her next action by framing it as a question. She then proceeds with the action and puts the slipper on the infant’s foot (Figure 3).

Mother:Wollen wir deine Schuhe wieder anziehen, hä?(Shall we put your shoes back on, yes?)

**Example 5.** VP10_4T (00:06:05.958–00:06:07.997)

The mother holds up the bottle of oil, and while unscrewing it, she announces her next action, namely to pour some massage oil into her hand (Figure 4).

Mother:Jetzt nehm’ ich noch ein bisschen Öl wieder.(Now I take a little bit of oil again.)

Interestingly, in these Examples 4 and 5, the verb might precede the actual action, but it is being uttered while preparing for that action. In this way, the mothers in Examples 4 and 5 set the stage for the subsequent action.

##### Subsequently Commenting on the Action

In a third pattern of synchrony, the action takes place first and is then followed directly by the utterance of the verb. In this case, the verb marks the end of the action.

**Example 6.** VP08_4T (00:02:09.162–00:02:10.967)

While dressing the infant, she accidentally pinched him. Directly after that, verbalizes her action (Figure 5).

Mother:tschuldigung, jetzt hab ich dich gezwickt.*(Sorry, I just pinched you.)*


##### Subsequently Commenting on the Infant’s Action (Infant Led)

Further analyses revealed another common pattern which consisted of the mother describing an action that the infant has performed. In this case, the infant was the actor providing the context and the mother was describing or elaborating on the action. In such a pattern, the verb is essentially not absolutely synchronous with the action, because the mother has to notice an action and then use a verb to address it. Interestingly, however, the verb is coupled with the infant’s own bodily experience. This presents the infant with words and meanings corresponding to his or her actions as in the following example.

**Example 7.** VP13_4T (00:01:10.502–00:01:11.833)


*While the mother is dressing the infant, she sees that he is putting the cream tube into his mouth. She then names the action.*


Mother:Die schmeckt gut, ne?(It tastes good, doesn’t it?)

**Example 8.** VP14_4T (00:03:21.889–00:03:23.396)

The infant has grabbed his clothes and is putting them in his mouth. The mother reacts to this action and pulls them away while using a negative statement with the verb “take.” In that moment, she is enacting the negation (Figure 6).

Mother:Nicht in den Mund nehmen!(Don’t take it in your mouth!)

##### Verbs Temporally Offset to Actions

Another observation was that the occurrence of these patterns varied in timing. In some cases, the verb and ongoing action co-occurred in time, or the announcement of the action and the action itself were coordinated very closely, as in Examples 1–8 presented above. In other cases, the temporal window connecting the verb with the action was longer, with either the action or the verb being delivered in a somewhat delayed manner. This observation led us to distinguish the usage of verb–action coordination into what we called *synchronous* or *temporally offset*. This distinction was motivated by the idea that two events spaced more widely apart might be difficult for the infant to associate.

In their behaviors, mothers also used verbs that were not perfectly timed with the ongoing action. We found cases in which mothers were talking about some aspect of the diaper changing routine, but there was no obvious connection to the ongoing action.

##### Decontextualized Verbs

Finally, we also observed cases in which the mothers used verbs referring to mental states or actions that were detached spatially or temporally from the ongoing interaction. We called this kind of verb use *decontextualized*, because no perceivable connection between the meaning of the verb and the activity taking place was apparent at that particular moment, as illustrated in the example below.

**Example 9.** VP08_4T (00:00:54.439–00:00:56.369)

While changing the infant’s diaper, the mother talks about the child’s day: She refers to actions and locations that are not part of either the ongoing interaction or behaviors. 

Mother:Und dann warst du draußen mit Oma?(And then, you were outside with grandma?)

In sum, our observations reveal much variability in the possible ways of associating language and action in the data. In some cases, the meaning of the verbs uttered was constrained by the ongoing action. Verbs are presented from the perspective of the child or describe upcoming or completed actions for the infant. In these cases, both the redundancy and the concreteness of the verb meaning are maximized. However, we also found other cases in which the actions and the verbs show no connection with each other or the verbs used are detached from the here and now of the ongoing interaction and therefore appear to be decontextualized.

#### 2.3.2. Quantitative Analysis

The case analyses led to the development of a coding scheme for all verbs occurring in maternal speech (see Table 1). The codes in the coding scheme are mutually exclusive. The mothers’ speech was transcribed using the software PRAAT and the utterances containing verbs were imported in ELAN (EUDICO Linguistic Annotator). This created annotations of all points in the video in which verbs were used and we could then view these segments together with the video and allocate them to one of the codes for verb–action coordination. To distinguish between the synchrony and temporally offset categories, we decided to use a time window of 2 s. Any verb–action coordination separated by more than 2 s was coded as *temporally offset*. This decision was motivated by evidence suggesting that infants can tolerate some degree of asynchrony [16], and that they can also perceive two events as contingent when they are slightly separated in time [31]. In addition, when we narrowed the window to one second, only 5% of the data changed.

To examine whether the coordination of verbs with action was modulated by infants’ gaze, we included existing coding of infant gaze locations in the analysis. Using a data modification module included in ELAN, we could create new annotations from the overlap of the various verb categories with infants’ gaze toward their mother or gaze away.

Having concluded the data preparation as described above, we created multiple measures which we then subjected to statistical analyses. More specifically, we calculated the proportion of verbs that was coordinated with action in all the different ways presented in Table 1. For this, we took the number of verbs in a specific category and divided it by the total number of verbs. 

Furthermore, we wanted to derive a measure representing the extent to which infant gaze modulated the behavior of the mother. To do this, we took the number of intervals of a given action–verb type overlapping with infants’ gaze at and divided it by the total number of verbs of this type. We then averaged these proportions over all dyads.

## 3. Results

Our data corpus consists of 1007 verbs in total used by the mothers in our sample. In our data, the interactions varied in duration (*M* = 7 min; 51 s, *MIN* = 2; 28, *MAX =* 15; 26) and also the amount of speech provided by the mothers also varied (*M* = 3 min; 6 s, *MIN* = 0; 2, *MAX =* 7; 44). We therefore controlled for this variability by calculating types per speech minute and tokens per speech minute by dividing the number of types/tokens by the amount of time the mother had spoken. The descriptive statistics of the corpus are illustrated in Table 2.

We were then interested in the linguistic kinds of verbs used in our corpus. Table 3 lists the types of verbs we encountered. The most prominent category was that of the auxiliary verbs, which were used mostly in copular constructions to express location and property of objects. Modal verbs and abstract verbs were the next most frequent categories. The frequency of occurrence of abstract verbs is largely due to the fact that this category lumps together many different kinds of verbs: It includes mental verbs, but also verbs with referential meaning that cannot be made transparent within one action but can be constructed within the interaction such as *enjoy, distract, imitate*, or *hurt*. A category which occurred also often was motor/tactile: It includes verbs referring to infant movement, such as *kick* or *crawl*, but also movements performed by the mothers on the infant’s body, such as *tickle, pinch*, or *kiss*. A further category included the verb *to come* (German: *kommen*) in many variations. The interesting observation about the way in which this verb is used is that it mostly described the mother picking up or pulling up the child. It is interesting that the mothers are reversing the roles and are producing this verb from the perspective of the infant, i.e., as if the infant were moving toward the mother instead of the mother moving the infant towards her. The sensory verbs mostly included variations of the verb *look* (German: *gucken*)*,* and were mostly used in contexts in which the mothers wanted to attract and direct infants’ attention to objects or actions. Very rarely the mothers used other sensory verbs such as *smell* or *listen*. The next category mostly lumps together verbs describing the ways in which actions are executed, such as *shake* or *roll*, and the effects of actions, such as *open* and *close* or *fall*. Verbs specifying object exchange were also used very often as well as verbs specifying verbal exchanges. Finally, verbs describing hygiene and care activities were sometimes used and are likely to be a product of actions taking place within the particular context.

Having described the verb corpus, we were then interested in quantifying the qualitative observations described in the previous section. Hence, we sought answers to the three following questions: First, we wanted to analyze to what extent verbs were embedded in the ongoing interaction with the infants. To address this question, we identified all verbs used by the mothers and assigned them to one of the aforementioned verb–action coordination types (see Table 1). Figure 7 provides the distribution of the proportion of different verb–action types. With respect to the timing, the most frequent type was that of Synchrony, with 66% (*SD =* 22) of the verbs spoken by the mother being coordinated with actions within a narrow temporal window of 2 s.

Comparing the differences in occurrence of the verb-action types we conducted a one-way repeated measures analysis of variance ANOVA, which revealed a large significant effect, *F*(3, 11) *=* 27.994, *p <* 0.001, *η^2^ =* 0.884. Post hoc pairwise comparisons (see indications in Figure 2) revealed significant differences between the types. More specifically, the occurrence of Synchrony was significantly higher than all other types, Temporally Offset was used significantly more than Decontextualized, and No Connection occurred significantly more than Decontextualized. No significant differences could be found for Temporally Offset and No Connection. 

The next analysis involved the distribution of the different types of temporal coordination both within and beyond the two-second time window (see Table 1). In most cases of verb use (see Figure 8a), mothers were led by the action of the infant (48%, *SD* = 12). This was followed by the occurrence of verbs, with which mothers synchronized their own action (39%, *SD* = 9). These two types of coordination made up 87% of the verbs accompanied by action within a two-second time window and strongly suggest that the most verbs that were used by the mothers related to the infants’ and their own actions.

Comparing the occurrence of the different verb-action types within a two-second time window, we conducted a further one-way repeated measures ANOVA indicating a large significant effect of verb types, *F*(3, 11) *=* 32.668, *p <* 0.001, *η^2^ =* 0.899. According to post hoc pairwise comparisons (see indications in Figure 8a), Full Synchrony occurred significantly more frequently than Verb First and Action First. In addition, Infant Led as a type was observed significantly more often than Verb First and Action First. No significant differences were found between Full Synchrony and Infant Led, and between Verb First and Action First. These results suggest that Fully Synchrony and Infant Led were the most frequent verb types when regarding the coordination of verbs with actions within a narrow window of 2 s.

Finally, comparing the differences in occurrence of the verb-action types offset by more than 2 s, we conducted a one-way repeated measures ANOVA displaying a large significant effect of verb types, *F*(2, 12) *=* 13.978, *p =* 0.001, *η^2^ =* 0.700. Post hoc pairwise comparisons (see indications in Figure 8b) revealed significant differences, according to which Verb First was observed significantly more often than Action First. In addition, the type of Infant Led occurred significantly more often than Action First. No significant differences were found between Verb First and Action First. These results suggest that, when a verb was provided offset, it was a response to infants’ action or the verb was an announcement of an action to be performed. 

### 3.1. Verb–Action Synchrony as a Function of Infant Gaze

Our second question was whether the mothers performed their multimodal behaviors in accordance with infant’s gaze. To address this, for each dyad, we contrasted the proportion of verb-action types used when the infant was gazing at the mother versus the proportion of verb-action types used when the infant was gazing away. As already described in the Method Section, we took the number of intervals of a given action–verb type overlapping with infants’ gaze at and divided it by the total number of verbs of this type. We then averaged these proportions over all dyads. 

The types Synchrony and Temporally Offset, both involving a temporal coordination between the action and the verb spoken by the mother, were found to occur in dependence on whether the infant attended to the mother or not (see Figure 9). Intriguingly, in those cases, in which there was no obvious connection between the verb and the occurring action, the contrast seemed less pronounced.

A one-way repeated measures ANOVA confirmed our observations by revealing a significant large effect for verbs being provided in accordance with infants’ gaze, *F*(1, 12) = 10.423, *p* = 0.007, *η*^2^ = 0.465. We found no interaction effect for the gaze and synchrony relationship. Post hoc pairwise comparisons revealed significant differences for Synchrony, *t*(13) = 5.709, *p* < 0.0001 and Temporally Offset coordination, *t*(13) = 3.553, *p* = 0.004. In both cases, the mothers used these types more when the infant was gazing at them than when the infant was gazing away. The differences were not significant for No Connection between verb and action, *t*(13) = 0.243, *p* < 0.812, or for Decontextualized verbs, *t*(12) = 0.457, *p* = 0.656. These results suggest that the two types of verbs, namely Synchrony and Temporally Offset, occur more frequently at just the moment when the infant was gazing toward the mother. The occurrence of other verb types, No Connection and Decontextualized, was less regulated by the infants’ gaze.

Next, we looked at whether infant gaze direction modulates mothers’ use of a narrow or loose temporal coordination of verbs with actions. The assumption for this analysis step was that different verb types will be pronounced within a narrow (within 2 s) in comparison to loose (spaced apart by more than 2 s) temporal coordination. For this purpose, we calculated the proportions for each dyad with the above-elaborated procedure and then averaged them over dyads. 

For a narrow coordination within a 2 s window, a one-way repeated measures ANOVA indicated a large significant effect for infant’s gaze, *F*(1, 13) *=* 15.142, *p* = 0.002, *η^2^* = 0.538, and a large effect for type of coordination, *F*(1, 13) *=* 5.200, *p* = 0.040, *η^2^* = 0.286, but no interaction effect for infants’ gaze and type of coordination (see Figure 10a). Post hoc pairwise comparisons revealed significant differences for Full Synchrony, *t*(13) *=* 3.225, *p* = 0.007, suggesting that, a higher proportion of verbs and actions were fully synchronized in time when infants were gazing toward their mother rather than when infants were gazing away. We found no significant comparisons for the other verb–action types. This result suggests that within a tight coordination, only fully synchronized verbs are systematically provided according to infants’ gaze direction.

For a loose coordination, spaced apart by more than 2 s, we also conducted a one-way repeated measures ANOVA and found a large significant effect for infant’s gaze, *F*(1, 13) *=* 10.673, *p* = 0.006, *η*^2^ = 0.451 (see Figure 10b). There was no interaction effect for infants’ gaze and type of temporal coordination. Post hoc pairwise comparisons revealed significant differences for Action First, *t*(13) *=* 2.736, *p* = 0.017, and Infant Led verbs, *t*(13) *=* 2.417, *p* = 0.031. These results suggest that even when the coordination was spaced apart, mothers still seemed to be sensitive to the gaze of their infant and used verbs to comment on concluded actions. In addition, the verbs frequently referred to the infants’ actions. 

### 3.2. Verb–Action Synchrony as a Predictor of Later Language Development.

Our third question addressed the relationship between the way verbs are embedded in the ongoing action and children’s later reported language development. We examined this question by obtaining data from the ELFRA-2 questionnaire and extracting the number of verbs in the toddlers’ productive vocabulary when they were 24 months old. We visualized the scores in a bar plot to explore the variability in the data. As illustrated in Figure 11, in our sample, there were children who at 24 months were reported to speak almost 40 verbs and there was one child who was reported not to produce a single verb.

We then conducted a linear regression analysis with infants’ number of spoken verbs at 24 months as the dependent variable and the different types of verb–action coordination as independent variables. We hypothesized that mothers who coordinate their use of verbs in time with the ongoing actions scaffold their children’s verb acquisition by providing them with tangible relationships between verbs and actions to a greater degree than mothers who provide the verb and action spaced apart. In the latter case, it could become more difficult for infants to associate the words to the actions. Results showed that verbs in such types as Full Synchrony, Temporally Offset, and Decontextualized made significant contributions. When taken all together, they predicted 70% of the variance in the number of infants’ spoken verbs at 24 months, *F*(4, 13) = 8.649, *p* < 0.01. More specifically, whereas Full Synchrony made a significant contribution (*β* = 0.619, *p* < 0.01)*,* the occurrence of Temporally Offset verbs made a significant but negative contribution to the model (*β* = −0.538, *p* < 0.01), indicating a negative relationship with later verb acquisition. A significant contribution to the model was also found for Decontextualized verbs (*β* = 0.675, *p* < 0.05). Finally, verbs of the type No Connection did not make a significant contribution in the model (*β* = −0.273, *p* = 0.238). These results suggest that, whereas the amount of spoken verbs not connected temporally and semantically to action (category “no connection” in Table 1) does not seem to relate to children’s verb acquisition, all other coordination forms of verbs—bearing a temporal or semantic (as was the case for decontextualized speech) connection with the ongoing actions—are related to children’s later verb production development. 

We further explored the role of mothers’ sensitivity to the gaze direction of their infant when embedding verbs in the ongoing action in a second linear regression model. Our hypothesis was that the infant gaze at the mother would further enhance the synchronous presentation of verbs and action by providing even richer multimodal experiences for verb learning. However, this analysis did not yield any significant effects.

Finally, to explore whether the use of verbs alone was associated with the infants’ acquisition of verbs, we ran correlations between the descriptive input measures (see Table 2) and infants’ number of spoken verbs as our outcome measure. None of these correlations were significant (tokens per speech minute: *r =* 0.198, *df =* 14, *p =* 0.498; types per speech minute: *r =* 0.115, *df =* 14, *p =* 0.695; type-token-ratio: *r =* 0.207, *df =* 14, *p =* 0.477).

## 4. Discussion

In the current study, we aimed at exploring caregivers’ use of verbs in everyday interaction and the ways in which these verbs are coordinated with ongoing actions. Departing from the assumption that experience of actions may guide the development of verb meaning [2], we set out to investigate the ways in which verbs in the maternal input are coordinated with mother’s own and/or her infant’s actions in early interaction. Assuming that social interaction supports learning by packaging information in ways that match infants’ perceptual mechanisms [32,33], we wanted to examine the patterns in this packaging to see whether these are provided according to infants’ gaze and whether they can be used to predict later language acquisition. We addressed these questions by analyzing naturalistic data from mother–infant interactions during diaper changing when the infants were six months old and related these data to infants’ reported verb production at the age of 24 months.

We initially collected and analyzed individual cases of verb use and observed that mothers temporally coordinate verbs with ongoing actions. The patterns of coordination ranged from a verb being presented in a fully and timed overlap with the ongoing action (Examples 1–8); verbs and actions connected to each other but provided temporally spaced apart; to verbs referring to actions from spatially, temporally, and semantically distant contexts (Example 9). Zooming in, we observed that mothers often designed their verbal and nonverbal behaviors to be perfectly synchronous (Examples 1 and 2). These observations confirm previous work [29] in which we suggested various types of synchrony. Further analyses revealed that mothers provided the verbs as a response to their infant’s action, thereby describing what the infant was doing (Example 8). This observation is consistent with the extensive literature on maternal responsiveness and language development (e.g., [26,27]). These qualitative observations surely provide evidence for the argument that the way in which infants’ experience the meaning of verbs in social interaction is very concrete, constrained to the here and now of their interactions and to their bodily experience. This finding contradicts literature suggesting that learning verbs is difficult because of the fleeting, ephemeral nature of the actions to which they refer (e.g., [6,34]). 

Expanding on these initial qualitative observations, we developed a scheme to code all verbs in our corpus according to type of verb–action coordination. We found that to a great extent (69%), the verbs spoken by the mother were coordinated with action within a narrow temporal window of 2 s. Coordinated within this tight window, most of the verbs were in response to the infant’s action followed by verbs presented in synchrony with the mother’s actions either on objects or the infant’s body. These two categories comprised 87% of the verbs coordinated with action within a two-second time window suggesting that early in development, the use of verbs is impressively narrowed down to very constrained meanings: The verbs refer to infants’ actions and describe actions of the mother that are either visible to the infant or actually enacted on the infant’s body. Similar observations have been reported recently by [35] for touch–language synchrony in nouns with older children. This finding was further supported by a corpus analysis of the types of verbs the mothers used in these interactions. This analysis revealed that only 20% of the verbs used were modals and abstract. The remaining 80% of verbs potentially afforded a synchronous presentation of their referential meaning. Among these verbs, 23% were copular constructions of the verbs *to be* and *have* expressing location or properties of objects and actions. The remaining 57% included verbs referring mostly to perceivable aspects of the interactions, namely, verbs specifying infants’ and mothers’ actions, spatial constructions, verbs describing the handling and exchange of objects. The referential meaning of these types of verbs could be enacted and made transparent to the infant though multimodal coordination. Arguably, our finding can be explained by the interactive context chosen for the data collection. It is plausible to assume that in a different activity that does not include infant care and hygiene, the same mothers would use different verbs and maybe coordinate them differently with actions. However, given the primary nature of routines such as feeding, bathing, and changing diapers, it is also reasonable to assume that such recurring activity contexts are most prominent for infants’ early experiences. Further research can pursue the question of how the verb types differ when provided within different activity contexts to give more insights into how particular activities (more or less routinized) are shaping social interaction. 

We should note that in our work, we did not analyze onomatopoeic expressions accompanying actions that mothers performed, because we focused on verbs as a specific word class. However, there are many instances of use of such sound symbolic relationships in our data and given the facilitative effect of sound symbolism for word learning [36,37], for our future work, it will be interesting to consider their role in scaffolding the word meaning in general.

In addition to these very concrete usages of verbs, we also found that a proportion of verbs was provided in a loose temporal coordination with action (outside a temporally narrow window of 2 s). Even fewer verbs showed no connection with the ongoing action suggesting that there was some temporal coordination but no semantic link to it. Finally, mothers also used few verbs that did not refer to the ongoing interaction (decontextualized verbs).

Our quantitative analyses further revealed that mothers coordinated these verb–action packages with their infant’s gaze. Comparing the proportion of verbs provided when the infants were gazing at the mother versus those when they were gazing away, we found a significant effect of gaze suggesting that a larger proportion of verbs were provided in accordance with infants’ gaze. Interestingly, differences were more accentuated for those verbs that were coordinated with the ongoing action than verbs showing no connection with the action or decontextualized verbs. This suggests that mothers considered their infants’ visual attention when using verbs with more direct reference to the ongoing activity than when using verbs that did not refer to the here and now. These findings provide the first insights into the nature of verb-action coordination as it becomes visible in early interactions with infants. 

Finally, we asked whether these coordination patterns predict infants’ later verb production. Therefore, we related the data on different patterns of verb–action coordination to data from follow-up parental reports when the infants were 24 months old. Following [25] indicating a relation between early speech and action presentation and later verb acquisition, we hypothesized that if social interaction and, more specifically, the coordination of verbs with actions is a factor scaffolding verb acquisition, then the variability in the use of maternal patterns should predict the variability in the production of verbs as captured by parental report at 24 months. The regression model confirmed our hypothesis. We found that 70% of the variance in infants’ spoken verbs could be predicted by the way in which mothers coordinated their verbs with actions. All verb–action types contributed significantly to the model except for the use of verbs showing no semantic connection to the action. Looking at the contributions of the individual variables to the model, we found that a tight synchrony of verbs and actions showed a positive relationship, whereas a temporal coordination that did not occur within a two-second time window made a negative contribution to the model. This suggests that a delayed or more distant presentation of the verb in relation to the ongoing action bears a negative relation to the children’s vocabulary at 24 months. Finally, we found that the use of decontextualized verbs made a positive contribution to the model suggesting that mothers who made more extensive use of this type of verb–action coordination had children who had more verbs in their productive vocabulary at 24 months or that children with a larger verb repertoire in the vocabulary had mothers who used more decontextualized verbs.

Contrary to our assumption, results did not support the hypothesis that verbs provided in accordance with infants’ gaze would relate significantly to the children’s later language learning. We can think of two possible explanations for this result: 

First, this might be due to studying infants at the age of six months. At this age, Nomikou and colleagues [28] report that mothers are very well synchronized with their children and efficient in providing their input: Because infants are interested in the physical environment and look away a lot, mothers were observed to monitor their infants’ attention more closely and to provide their input exactly when the infant is looking. In contrast, three-month-olds spend a lot of time just gazing at their mothers. For the data presented here, this can mean that splitting the verbs into the two categories (those provided at the moment of infants’ gazing and other) does not reflect the nature of the interaction. A more crucial behavior—that we did not consider here—might be related to the objects. For example, naming an action with an object while the infant is attending to it could probably start becoming of significant importance for verb development from the age of six months on. Further analyses should thus include verb–action synchrony occurring with other locations at which infants gaze, such as gaze toward objects. 

Second, the lack of a relationship between verbs being provided in line with infants’ gaze and their later language learning could be due to the nature of verbs as building blocks of grammar which are relational and thus inherently afford less one-to-one mapping. Instead, as a unit of input, they are embedded within clauses. Thus, it is possible that the mothers started a clause when their child was looking, but at the moment when the verb was mentioned, the child was looking away. To disentangle this, future analyses could attempt sequential measures which would be able to provide more details on the exact order and timing of behaviors.

In sum, our results revealed that taking all the verb types into account rather than splitting them according to the infants’ attention for a very short moment is a more powerful predictor of the children’s reported language development. Our results further speak for the existence of semantic contingency already recognized in [27] as we found that the Full Synchrony and Infant Led types took up the biggest proportion of verbs used. These relate to the mother’s own action or the infant’s actions, making use of the tactile modality.

## 5. Conclusions 

In this article, we explored the different ways of how mothers synchronize their verbs with actions in the early input to six-month-old infants. Using qualitative and quantitative methods, we found that the verbs used by mothers in these early interactions were tightly coordinated with the ongoing action and very frequently responsive to infants’ actions. In addition, we could observe that when using verbs with more direct reference to the ongoing activity than when using verbs that did not refer to the here and now, mothers’ input was more tuned to infants’ visual attention.

The different ways of verb-actions-coordination could significantly predict the number of spoken verbs in children’s vocabulary at 24 months. We can therefore conclude that all verb–action types contributed significantly to the model except for the use of verbs showing no semantic connection to the action. Verbs and action spaced apart by more than 2 s were negatively related to the children’s reported vocabulary.

In sum, our finding that most verbs in maternal input are presented in a tight temporal relationship to either the mother’s or the infant’s action combined with the finding that the verbs in the early input are predictive of later verb development support the importance of social interaction. They show that it is a crucial source of sensorimotor experience and should be taken into account in future studies investigating the cognitive foundations of verb learning and proposing suitable components for necessary representations.

## Figures and Tables

**Figure 1 brainsci-07-00052-f001:**
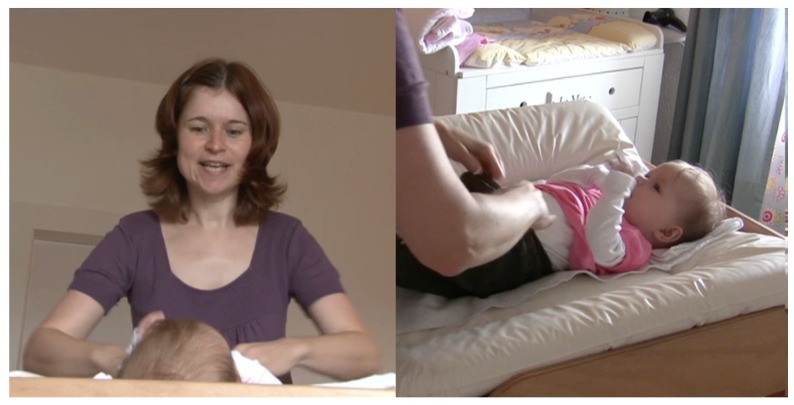
Camera setup.

**Figure 2 brainsci-07-00052-f002:**
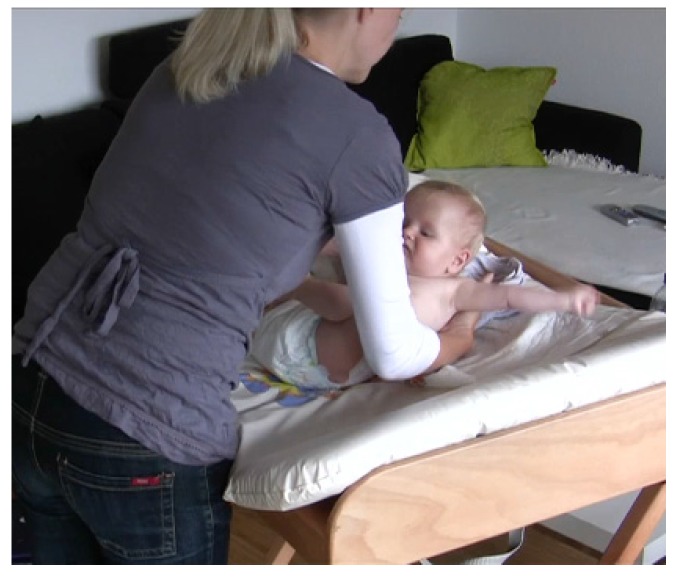
Mother accompanying the lifting movement of the infants’ body with her utterance.

**Figure 3 brainsci-07-00052-f003:**
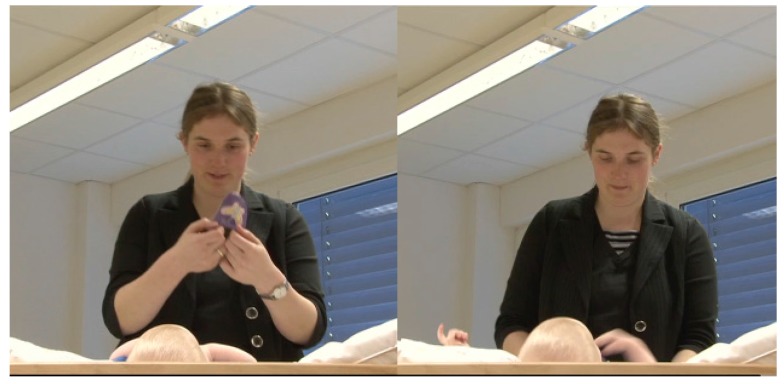
A mother announcing her next action that is performed on the infant.

**Figure 4 brainsci-07-00052-f004:**
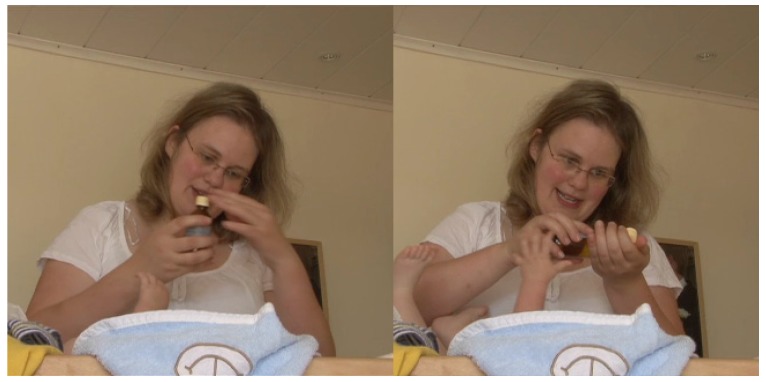
A mother announcing her next action that is performed on an object.

**Figure 5 brainsci-07-00052-f005:**
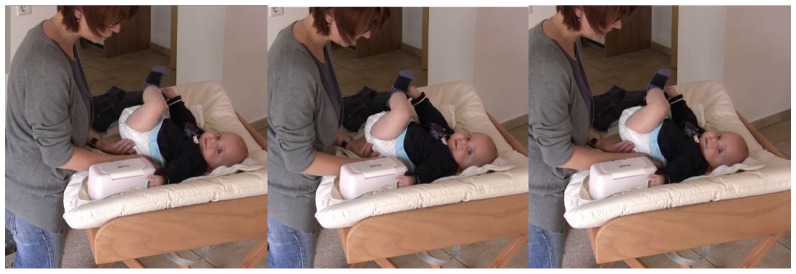
A mother subsequently commenting on her action.

**Figure 6 brainsci-07-00052-f006:**
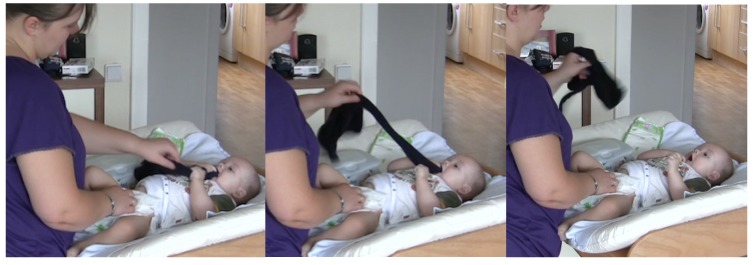
A mother commenting on the infants’ action.

**Figure 7 brainsci-07-00052-f007:**
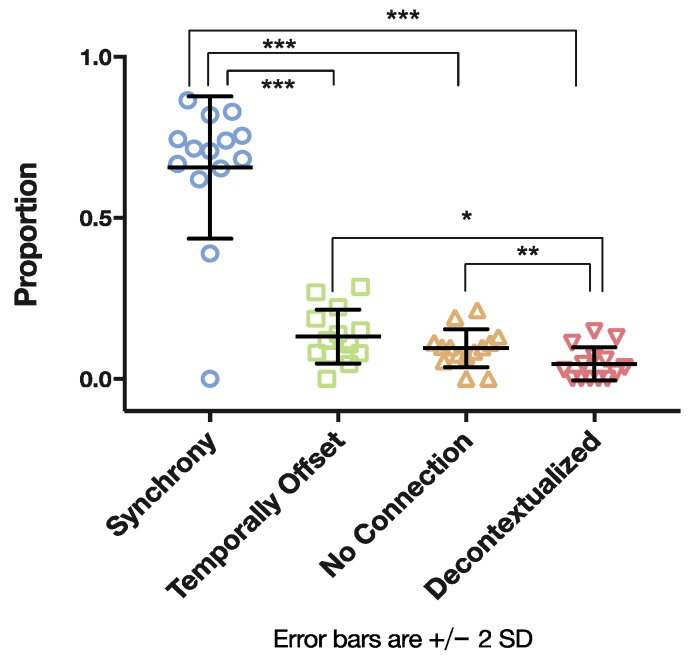
Verb–action categories (* *p* < 0.05; ** *p* < 0.01; *** *p* < 0.001).

**Figure 8 brainsci-07-00052-f008:**
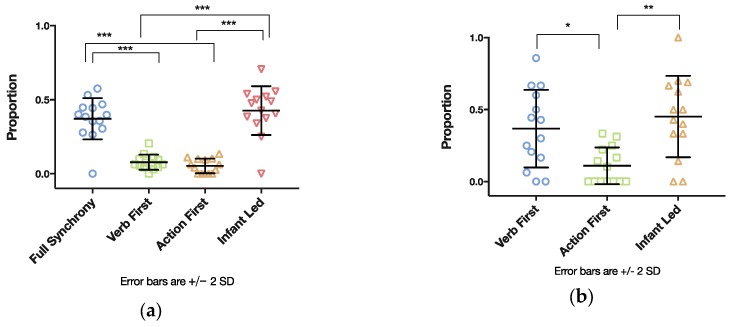
(**a**) Types of temporal coordination within a two-second time window. (**b**) Types of temporal coordination offset by more than 2 s (* *p* < 0.05; ** *p* < 0.01; *** *p* < 0.001).

**Figure 9 brainsci-07-00052-f009:**
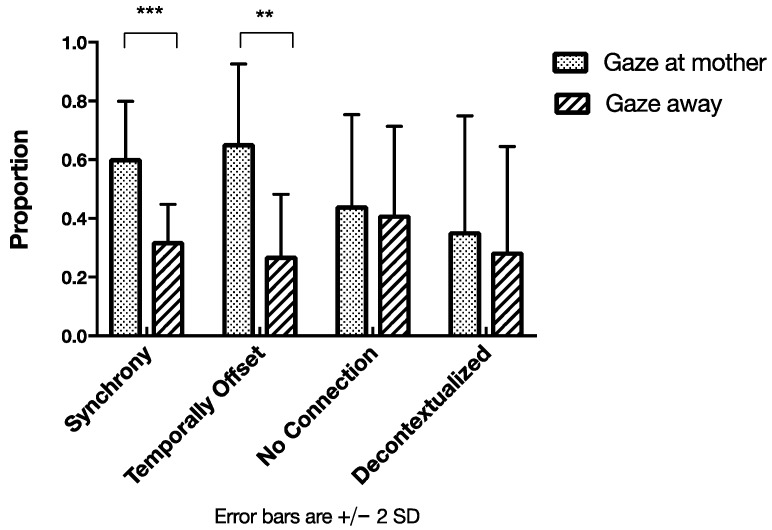
Types of verb–action coordination overlapping with categories of infant gaze (* *p* < 0.05; ** *p* < 0.01; *** *p* < 0.001).

**Figure 10 brainsci-07-00052-f010:**
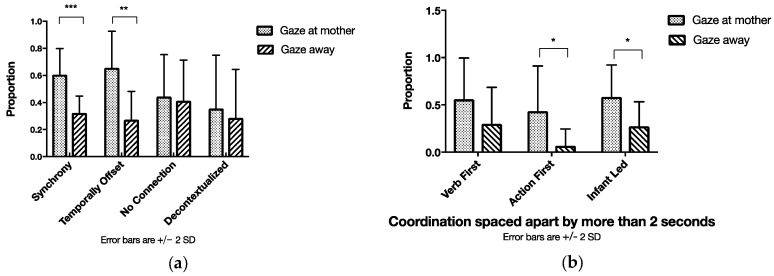
(**a**) Types of temporal coordination within a two-second time window overlapping with categories of infant gaze. (**b**) Types of temporal coordination offset by more than 2 s overlapping with categories of infant’s gaze (* *p* < 0.05; ** *p* < 0.01; *** *p* < 0.001).

**Figure 11 brainsci-07-00052-f011:**
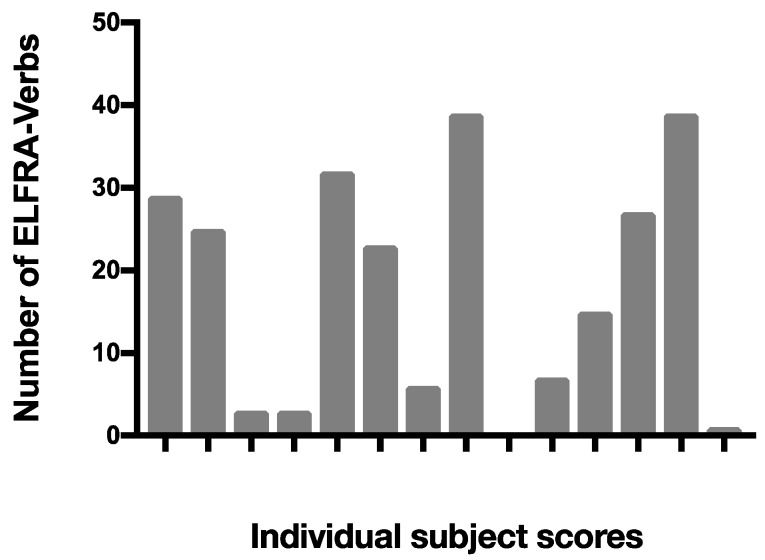
Infants’ number of spoken verbs at 24 months as reported by their parents.

**Table 1 brainsci-07-00052-t001:** The coding schema.

Code		Description of Verb–Action Overlap
Synchrony	Full	Complete synchrony
	Verb first	Action follows verb within 2 s
	Action first	Verb follows action within 2 s
	Infant led	Verb as a response to infant action. Action follows verb within 2 s
Temporally offset	Verb first	Action follows verb after 2 s or more
	Action first	Verb follows action after 2 s or more
	Infant led	Verb as a response to infant action. Action follows verb after 2 s or more
No connection	No connection	Not visible between verb and action. Verb refers to the interaction
Decontextualized	Decontextualized	Verb does not refer to the current situation (but to spatially and temporally distant events)

**Table 2 brainsci-07-00052-t002:** Verbs in Input (M refers to the mean value, SD to the standard deviation, MIN and MAX to minimum and maximum respectively.

	M	SD	MIN	MAX
Tokens per speech minute	22.80	8.83	0	36.39
Types per speech minute	10.63	5.89	0	23.45
Type-Token-Ratio	0.46	0.14	0	0.69

**Table 3 brainsci-07-00052-t003:** Types of verbs in the input.

Type	Frequency in %	Examples
Auxiliary/Copular	27	*be, have*: mostly copular constructions but also used to express existence and property
Abstract	10	*believe, know, forget, hope*; but also *need, be worth it, imitate*
Modal	10	all German modals: *dürfen* (to be allowed), *können* (to be able to), *mögen* (to like), *müssen* (must, to have to), *sollen* (should, to be supposed to), *wollen* (to want).
Motor/Tactile	9	*crawl, kick, stand up*; but also *clap, kiss, tickle*
Come	7	expressing movement towards the mother, performed by mother and/or infant
Sensory	7	*look, browse, see*; but also *hear, smell*
Spatial	7	any spatial construction such as: *put on*, *pull over*, *wrap around*, *fall into*, *take out*
Manner/Change of state	6	*shake*, *turn*, *pull*, but also *open* and *close* mostly in the context of handling objects or moving body parts
Do	6	in German, these verbs are combined with prepositions similar to the way the verb *“put”* is used in English
Object exchange	6	*give* and *take*
Care	4	*dress, eat, sleep, pee*
Verbal Behavior	3	*talk, sing, tell, squeak*

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
