# Peer review of "Verbs in Mothers’ Input to Six-Month-Olds: Synchrony between Presentation, Meaning, and Actions Is Related to Later Verb Acquisition"

_brainsci, 2017, doi:10.3390/brainsci7050052_

Round 1

Reviewer 1 Report

I found a lot to like about this paper; I think descriptive analytical work of this sort is important.  My concerns are mainly about the analyses I found hard to follow from their description.  To my mind the authors' task in revision is really only to make more clear what is in the analyses.  Other than that I found the paper fully adequate.  Many of my comments are fairly minor in nature and should be easy to improve, but I do think that these sorts of improvements are necessary.

There were 14 infants.  But how many observations per infant?  (i.e., how many verb tokens were coded?)  I couldn't find this in the paper, but it's essential information.

I did have some questions about the time window.  +/- 2 sec. around when a word is used did not seem to me to be a narrow window at all.

A whole multiword sentence easily could be 4 seconds.  (It was also not clear on p. 7 whether the 2s window is around the onset of the word, or the midpoint of the word, or its offset, or what.)  If the mother says, "Die schmeckt gut, ne?" (line 226) the 4-sec window probably includes the whole utterance.  Are the results similar if a narrower window is used?  I could imagine that the correlations to later development would be stronger with a narrower window.  Or not. But a 4-second window is not precise synchrony.

Is there reason to believe that the diaper changing context is

typical?  I do not know.  Perhaps the language used at this time is unusual in some respects, possibly because the routine is so

unvariable from one time to the next.

== minor comments

the legends on some of the bar charts were quite small.

line 95: it is a quirk of the word "timely" that it can be an

adjective but not an adverb; it is not felicitous in this sentence.

Figure two is labeled as "percentage" but the numbers are

proportions.  (also 4 decimal places is probably beyond the accuracy of the data)

The initial bar graphs don't convey much information (eight numbers per plot: 4 means and 4 SDs).  Perhaps a plot of each mother's means per category could be overlaid on the bars?  (I leave this entirely up to the discretion of the authors though.)

I had some doubts about identifying "maternal responsiveness" as

"things the mother does while the infant is looking at her" (section 3.1).  Something seems a little off in the terminology.  A response is usually a reaction to an event.  If the infant looks at his mother for 10 seconds, and in the 8th second the mother says something, is this "responsiveness"?  I don't know.  I'm not doubting the relevance of this measure (it's very interesting), I'm just concerned that readers will find this part harder to follow given the way things are described.  A less theory-dependent section header would be "Action-verb synchrony as a function of infant gaze to the mother" rather than the more opaque "Action-verb syncrony [sic] as a form of maternal responsiveness."

For some of the plots I couldn't work out what the denominator was in the proportions.  In 5b, for example, the "gaze at" proportions don't sum to one, nor do the "gaze away" proportions... what is the calculation?  I think I know the numerator in each fraction, but I can't tell what the denominator is.

In many cases the results are reported as differences but the

direction isn't explicitly stated.  I (and possibly others) have a

preference for stating the direction.  For example, line 345 says

"Post-hoc pairwise comparisons revealed significant differences for the cases in which the verb was used to conclude the action t(13) =2.736, p = .017, and the cases in which the mothers followed theaction of the infant and provided a label for it t(13) = 2.417, p =.031 suggesting that even in the cases in which the coordination was spaced apart the mothers were still sensitive to the gaze of the infant."  "Differences" leaves open the direction.  Why not say "When the verb concluded the action, mothers were significantly more likely to... "?  (I can't actually fill in the example, because I honestly don't know what is being reported here; see previous comment.)

Figure 6 does not have an x axis in my copy of the paper.

line 365 - what does "percentage of spoken verbs" mean?  Is it the

number of verbs the child is reported to know, divided by the total number of words the child is reported to know?  Is total number of words also predicted by the same variables?  The discussion (line 462) says "spoke more verbs" but this is not what the analysis seems to report (i.e., not the same thing as *proportion* of verbs).

Author Response

Thank you very much for your comments. We hope to have addressed everything.

Please find our answers in red.

I found a lot to like about this paper; I think descriptive analytical work of this sort is important.  My concerns are mainly about the analyses I found hard to follow from their description.  To my mind the authors' task in revision is really only to make more clear what is in the analyses.  Other than that I found the paper fully adequate.  Many of my comments are fairly minor in nature and should be easy to improve, but I do think that these sorts of improvements are necessary. 

We have added parts in the analysis explaining our procedure. We hope that this part is clearer now.

There were 14 infants.  But how many observations per infant?  (i.e., how many verb tokens were coded?)  I couldn't find this in the paper, but it's essential information.

We have included a part about verb type and tokens in the revised version. Also some descriptive information about the types of verbs.

I did have some questions about the time window.  +/- 2 sec. around when a word is used did not seem to me to be a narrow window at all.

A whole multiword sentence easily could be 4 seconds.  (It was also not clear on p. 7 whether the 2s window is around the onset of the word, or the midpoint of the word, or its offset, or what.)  If the mother says, "Die schmeckt gut, ne?" (line 226) the 4-sec window probably includes the whole utterance.  Are the results similar if a narrower window is used?  I could imagine that the correlations to later development would be stronger with a narrower window.  Or not. But a 4-second window is not precise synchrony.

We are grateful for this comment. A two second window means that the verb was within this interval (1 second before and after). Even though the 2 second-window is better motivated by the available literature, following the reviewer’s suggestion, we coded thee dyads within a narrow window of 1 second (0.5 before and 0.5 after). As we mention in the revised version, the data changed only by 5 %.

Is there reason to believe that the diaper changing context is typical?  I do not know.  Perhaps the language used at this time is unusual in some respects, possibly because the routine is so unvariable from one time to the next.

Thank you for this question. We have provided some motivation for the choice of context in the methods section and we discuss it also in the Discussion section.

 == minor comments

the legends on some of the bar charts were quite small.

All graphs were modified

line 95: it is a quirk of the word "timely" that it can be an adjective but not an adverb; it is not felicitous in this sentence.

We have corrected it. In addition, the whole manuscript was proofread by a native speaker.

Figure two is labeled as "percentage" but the numbers are proportions.  (also 4 decimal places is probably beyond the accuracy of the data)

Corrected!

The initial bar graphs don't convey much information (eight numbers per plot: 4 means and 4 SDs).  Perhaps a plot of each mother's means per category could be overlaid on the bars?  (I leave this entirely up to the discretion of the authors though.)

Thank you for this suggestion, we have changed the graphs.

I had some doubts about identifying "maternal responsiveness" as "things the mother does while the infant is looking at her" (section 3.1).  Something seems a little off in the terminology.  A response is usually a reaction to an event.  If the infant looks at his mother for 10 seconds, and in the 8th second the mother says something, is this "responsiveness"?  I don't know.  I'm not doubting the relevance of this measure (it's very interesting), I'm just concerned that readers will find this part harder to follow given the way things are described.  A less theory-dependent section header would be "Action-verb synchrony as a function of infant gaze to the mother" rather than the more opaque "Action-verb syncrony [sic] as a form of maternal responsiveness."

Thanks for the suggestion! We can absolutely relate to your concern. We have modified according to the proposal.

For some of the plots I couldn't work out what the denominator was in the proportions.  In 5b, for example, the "gaze at" proportions don't sum to one, nor do the "gaze away" proportions... what is the calculation?  I think I know the numerator in each fraction, but I can't tell what the denominator is.

We have added a section explaining exactly how the proportions were calculated. We hope that is is clear now.

In many cases the results are reported as differences but the direction isn't explicitly stated.  I (and possibly others) have a preference for stating the direction.  For example, line 345 says "Post-hoc pairwise comparisons revealed significant differences for the cases in which the verb was used to conclude the action t(13) =2.736, p = .017, and the cases in which the mothers followed theaction of the infant and provided a label for it t(13) = 2.417, p =.031 suggesting that even in the cases in which the coordination was spaced apart the mothers were still sensitive to the gaze of the infant."  "Differences" leaves open the direction.  Why not say "When the verb concluded the action, mothers were significantly more likely to... "?  (I can't actually fill in the example, because I honestly don't know what is being reported here; see previous comment.)

We have included a description of the direction of the difference for all post-hoc tests.

Figure 6 does not have an x axis in my copy of the paper.

We changed this Figure from a scatterplot to a bar graph.

line 365 - what does "percentage of spoken verbs" mean?  Is it the number of verbs the child is reported to know, divided by the total number of words the child is reported to know?  Is total number of words also predicted by the same variables?  The discussion (line 462) says "spoke more verbs" but this is not what the analysis seems to report (i.e., not the same thing as *proportion* of verbs).

Changed the results to number of verbs not proportion throughout the paper.

Reviewer 2 Report

I am pleased to see a study that examines naturalistic interactions between mothers and infants. Studies that explore mechanisms of development in the everyday life of the infant, supplemented with experimental studies in the lab, are necessary to generalize lab findings. The careful development of the coding scheme, and examples provided, were helpful to understand the relation between observations and the study question being explored (I provide some suggestions for clarity of presentation below). The analyses revealed a relationship between temporal synchrony of action/verb presentation at 6 months and verb outcomes at 24 months. Although in a small sample, this finding suggests the importance of multimodal input to enhance learning in infants. The relationship between infant gaze and mothers’ synchrony was interesting, and suggests that infant gaze influences mothers’ responsive behavior; however, it might be enhanced by time series/sequential analyses.

Line 117: “multimodal practises” is unclear. Maybe just multimodal responses? Or behaviors?

Line 121: I would suggest a different word to replace “reckoned” maybe “surmised” or just “hypothesized”?

“…needs to be enhanced by being responsive to infants’ gaze in order to function as meaningful and rich input guiding learning processes.” Suggested change: “…and rich input that could guide learning processes.”

Line 129: “In a next step we related our findings with parental reports…” Suggested change: “In the next step, we related our findings to parental reports…”

Line 140: “…on a higher level” maybe change to “…at a higher level”

Line 153: “Out of the…we were able to collect the parental

Line 161: remove “s” from “attempts”

Line 167: “…at the types”

Lines 169-216: The section starts with “One involved either…” I suggest wording “One pattern involved either…” The pattern is in Example 1, but then there are other numbered examples to follow (are they also illustrating the first pattern?) and it is unclear where the 2nd pattern is (lines 194-195?)-this should be made clear. Then line 214 explicitly introduces the third type of coordination. Make it clear which is pattern 2 is and which example(s) illustrate it. Then line 216 “Finally, a very common practice…” so this is a fourth pattern of coordination? I would suggest in Line 169 you state “There were 4 patterns of coordination that varied relative to when the verb occurred relative to the action and whether or not the infant was involved in the action” (or something like this)

Line 236 “Another observation was that the…”-again, try to introduce how many patterns there were and how they varied before describing all of them and giving examples? (ditto for line 247)

Line 257: I like the summary of the observations, but I think it would be helpful for the reader to have some sense before reading all of the patterns and examples how they vary-then the examples are more clear in the reader’s mind as they read through them (in terms of how they vary relative to the study question). (similar to what is done in the Discussion, lines 400-406)

Lines 281-283: Instructions to the author seem to be mistakenly included in the manuscript: “This section may be divided by subheadings. It should provide a concise and precise description of the experimental results, their interpretation as well as the experimental conclusions that can be drawn.”

Line 292: delete “categories” that first occurs in the sentence

Line 293: 69% doesn’t seem to match the graph? I think that Figure 2 is just Figure 3 with different x-axis labels. Please provide the correct Figure 2.

Line 301: change “out” to “up”

Figure 3b: it is interesting that the same pattern of temporal coordination offset by more than two seconds follows temporal coordination within a two second time window. It would be interesting to know which verbs these were (and verb outcomes tested at a later timepoint) and how infants’ gaze (or other behavior) relates to which verbs are synchronous vs. offset (Fig 4 looks at gaze relative to synchrony vs. offset presentation, but specific verbs could be explored)

Line 307: “multimodal practices” again, change to ‘responses’ or ‘behaviors’ I think?

(if you prefer to keep the word practices, it is spelled with a ‘c’ here and an ‘s’ earlier in the manuscript)

ANOVA relating to Figure 4-gaze and synchrony. One interesting question for the verbs that are temporally offset would be to look at the timing of gaze relative to the presentation of the verb and /or action that is offset. Maybe if you look at sequential patterns/time series, you might see that a verb (or action) occurs when the infant looks. So, for example, the mother was only talking (verb utterance), but the infant looks, so she performs the action. The infant gaze ‘elicits’ the presentation of both, which in some cases makes it offset? But maybe this is still better than not being presented together at all?

Similarly, for Figure 5 and corresponding analysis, rather than gaze at/gaze away, with time series and sequential time information, you could see the timing of gaze relative to verb/action

I think in 5b and analyses you get at this question somewhat, but temporal analyses would possibly reveal more precise patterning/timing effects (very interesting!)

Line 374: delete extra “verbs” in the sentence

Lines 384-388: maybe temporal/sequential analyses would reveal something related to gaze and verb/action synchronous presentation and outcomes?

Line 390: “…aimed to explore…”

Line 424: “infants’” (apostrophe needed)

Line 438: “mothers” (no apostrophe)

Line 449: delete ‘ on families’

General question: I realize it is a small sample, but could you look at individual differences in mothers who show synchrony vs. decontextualized? (to address the comment, lines 461-462, “…mothers who made more extensive use of verbs had children who at 24 months…”)

Also, could you assess the verbs that mothers used vs. the ones that children know at 24 months? Even if not specifically learning the exact verbs mothers used, did infants learn more verbs that lend themselves to synchronous action/verb presentation (vs. more abstract verbs or ones that are less tightly synced to behavior)?

Author Response

We are very grateful for the careful and constructive comments. We hope that we have been able to address them. Please find our answers below in red.

I enjoyed this paper and found the work to be well done. It is an important question and findings are interesting. My comments are as follows:

I enjoyed reading this manuscript and the multi-method approach to analyzing infant-directed speech around verbs.  There are many positive features to this research. The question of how infants learn verbs—words that express abstract relations among objects, environment and people—has a long history. The research is theoretically grounded in the proposition that mothers engage in actions that are tightly coordinated to the verbs they use, and responsive to infant action. The literature review is well done and thorough, revealing the authors’ expertise on the social context of language development. The authors applied a very careful analysis to mothers’ infant-directed speech, and found support for this temporal coordination. Moreover, tight coordination during diaper-changing in infancy predicted infant verb learning at 2 years. This bottom-up, embodied approach to understanding early verb learning is admirable. The result that synchrony explained 68 % of the variance in infants’ percentage of spoken verbs at 24 months is quite impressive. I also found the focus on language in the context of diaper-changing to be ecologically valid, and well motivated. Camera set up was well done to capture the important actions of both participants.

Although the sample size is small (n=14), the micro-genetic and qualitative aspects of the study are valuable. The pictures with accompanying text examples of what mothers said as they enacted the diaper-changing process are rich. The grounded approach to developing a coding system and details around steps involved in that process are well executed (Table 1 presents a very clear coding system for temporal synchrony of words and action). I am highly in favor of our field venturing into different types of methods to advance theoretical ideas and empirically derived code systems as was done here. I do believe that there is insufficient attention to careful study of a few cases to reveal learning processes. The consequence is that too many studies rely on large samples but with diluted measurement and detail. 

I therefore have only a couple of suggestions for potential improvement. My positive evaluation of this paper, of course, assumes that the journal is open to small samples (n=14) and largely descriptive, microgenetic studies. If so, this is a commendable piece of research.

First, I would request a very explicit statement (in opening and then discussion) as to how precisely this work advances on what is known. The authors cite several excellent papers on verb learning, but do not come right out and say what is missing and how this rigorous analysis adds to the literature. As someone who engages in studies of word learning under naturalistic contexts, I appreciate the great effort that went into this work, but others may wonder what is new here and why.

We are grateful for this comment.

In the revised version, we have added a paragraph in the introduction and the discussion.

Second, I wondered whether there is a way to statistically test some of the descriptive data (particularly the first reported results where no statistics are used). At present, many results are based on bar graphs that display means and error bars without any statistical analysis. Of course, I recognize that a picture is worth a thousand words, but there might be some way of generating some analyses, such as using conditional probabilities to test if synchrony of sequences differs from chance occurrence or something (I have not thought this through, but it would benefit the paper).

We did run analyses and include them in the revised version. We also indicate in the graphs which differences are significant.

Under section 3.1, formal analyses begin to appear. However, there should be a bit more clarification on precisely what contrasts are being tested in these analyses (for instance, the authors should explicitly state they are comparing variable X under conditions of infant gaze versus no gaze within a dyad).

We have added a section in the Method-Analysis in which we describe all the measures we used. We hope that our method has become more clear now.

Third, the writing requires a bit of work in several places. There are several grammatical errors and awkward sentence constructions. The authors might ask for assistance in writing by a native-English speaker just to enhance readability. In a few places, entire sentences were hard to follow for their logic. For example, the opening sentence to Results states, “This section may be divided by subheadings.” I was left wondering what do the authors mean? That they are using subheads? Or not? And why is that something that matters?

thank you for this. We agree the initial submission was not optimal. We have addressed the point by revision our sections. 

In addition, a native speaker proofread our revised version.

Finally, some graphics can be improved. Results are well organized with accompanying visuals that present interesting descriptive data (e.g., the synchrony of mothers’ verbs and actions). However, I recommend the graphs be created with a program such as sigma plot or something else, as they currently look like they are taken from an SPSS printout. And, some choices of graphics are unclear—namely Figure 6’s use of a scatter plot for what I think are just mean proportion of verbs in each child’s vocabulary, which are perhaps better just individual bar graphs? Scatterplots suggest 2 variables are being related to one another, and I kept trying to find the x-axis values.

Thanks for this suggestion, we have changed all the graphs and they do look much better then before. We hope the reviewer agrees.

Reviewer 3 Report

I enjoyed this paper and found the work to be well done. It is an important question and findings are interesting. My comments are as follows:

I enjoyed reading this manuscript and the multi-method approach to analyzing infant-directed speech around verbs.  There are many positive features to this research. The question of how infants learn verbs—words that express abstract relations among objects, environment and people—has a long history. The research is theoretically grounded in the proposition that mothers engage in actions that are tightly coordinated to the verbs they use, and responsive to infant action. The literature review is well done and thorough, revealing the authors’ expertise on the social context of language development. The authors applied a very careful analysis to mothers’ infant-directed speech, and found support for this temporal coordination. Moreover, tight coordination during diaper-changing in infancy predicted infant verb learning at 2 years. This bottom-up, embodied approach to understanding early verb learning is admirable. The result that synchrony explained 68 % of the variance in infants’ percentage of spoken verbs at 24 months is quite impressive. I also found the focus on language in the context of diaper-changing to be ecologically valid, and well motivated. Camera set up was well done to capture the important actions of both participants.

Although the sample size is small (n=14), the micro-genetic and qualitative aspects of the study are valuable. The pictures with accompanying text examples of what mothers said as they enacted the diaper-changing process are rich. The grounded approach to developing a coding system and details around steps involved in that process are well executed (Table 1 presents a very clear coding system for temporal synchrony of words and action). I am highly in favor of our field venturing into different types of methods to advance theoretical ideas and empirically derived code systems as was done here. I do believe that there is insufficient attention to careful study of a few cases to reveal learning processes. The consequence is that too many studies rely on large samples but with diluted measurement and detail. 

I therefore have only a couple of suggestions for potential improvement. My positive evaluation of this paper, of course, assumes that the journal is open to small samples (n=14) and largely descriptive, microgenetic studies. If so, this is a commendable piece of research.

First, I would request a very explicit statement (in opening and then discussion) as to how precisely this work advances on what is known. The authors cite several excellent papers on verb learning, but do not come right out and say what is missing and how this rigorous analysis adds to the literature. As someone who engages in studies of word learning under naturalistic contexts, I appreciate the great effort that went into this work, but others may wonder what is new here and why.

Second, I wondered whether there is a way to statistically test some of the descriptive data (particularly the first reported results where no statistics are used). At present, many results are based on bar graphs that display means and error bars without any statistical analysis. Of course, I recognize that a picture is worth a thousand words, but there might be some way of generating some analyses, such as using conditional probabilities to test if synchrony of sequences differs from chance occurrence or something (I have not thought this through, but it would benefit the paper). Under section 3.1, formal analyses begin to appear. However, there should be a bit more clarification on precisely what contrasts are being tested in these analyses (for instance, the authors should explicitly state they are comparing variable X under conditions of infant gaze versus no gaze within a dyad).

Third, the writing requires a bit of work in several places. There are several grammatical errors and awkward sentence constructions. The authors might ask for assistance in writing by a native-English speaker just to enhance readability. In a few places, entire sentences were hard to follow for their logic. For example, the opening sentence to Results states, “This section may be divided by subheadings.” I was left wondering what do the authors mean? That they are using subheads? Or not? And why is that something that matters?

Finally, some graphics can be improved. Results are well organized with accompanying visuals that present interesting descriptive data (e.g., the synchrony of mothers’ verbs and actions). However, I recommend the graphs be created with a program such as sigma plot or something else, as they currently look like they are taken from an SPSS printout. And, some choices of graphics are unclear—namely Figure 6’s use of a scatter plot for what I think are just mean proportion of verbs in each child’s vocabulary, which are perhaps better just individual bar graphs? Scatterplots suggest 2 variables are being related to one another, and I kept trying to find the x-axis values.

Author Response

We thank the reviewer for the very constructive comments.

Please find our answers below in red 

I am pleased to see a study that examines naturalistic interactions between mothers and infants. Studies that explore mechanisms of development in the everyday life of the infant, supplemented with experimental studies in the lab, are necessary to generalize lab findings. The careful development of the coding scheme, and examples provided, were helpful to understand the relation between observations and the study question being explored (I provide some suggestions for clarity of presentation below). The analyses revealed a relationship between temporal synchrony of action/verb presentation at 6 months and verb outcomes at 24 months. Although in a small sample, this finding suggests the importance of multimodal input to enhance learning in infants. The relationship between infant gaze and mothers’ synchrony was interesting, and suggests that infant gaze influences mothers’ responsive behavior; however, it might be enhanced by time series/sequential analyses.

We agree with the reviewer that the relationship between infants’ gaze and mothers’ synchrony bears a lot of potential for further analyses. The focus of our paper, however, was to make a first step in exploring the types of synchrony and in identifying the patterns that are the most common in early interactions. Our results so far revealed that in dependence of the time window, some verb types can be further investigated. More specifically, we think that in future analysis, within a narrow time window of 2 seconds, sequential analyses can be applied to verbs that are provided in full synchrony. But outside this narrow window, as our analyses suggest, other verbs (e.g., Infant Led) might be interesting to further investigate. 

Since we already combine qualitative and quantitative methods, we decided that including more analyses would lead to a different paper altogether, or maybe overload it, as we would be reporting many different types of analysis. We hope that the reviewer will understand our decision to pursue this suggestion in our future work.

Line 117: “multimodal practises” is unclear. Maybe just multimodal responses? Or behaviors?

Done

Line 121: I would suggest a different word to replace “reckoned” maybe “surmised” or just “hypothesized”?

Done

“…needs to be enhanced by being responsive to infants’ gaze in order to function as meaningful and rich input guiding learning processes.” Suggested change: “…and rich input that could guide learning processes.”

Done

Line 129: “In a next step we related our findings with parental reports…” Suggested change: “In the next step, we related our findings to parental reports…”

Done

Line 140: “…on a higher level” maybe change to “…at a higher level”

Done

Line 153: “Out of the…we were able to collect the parental

Done

Line 161: remove “s” from “attempts”

Done

Line 167: “…at the types”

Done

Lines 169-216: The section starts with “One involved either…” I suggest wording “One pattern involved either…” The pattern is in Example 1, but then there are other numbered examples to follow (are they also illustrating the first pattern?) and it is unclear where the 2nd pattern is (lines 194-195?)-this should be made clear. Then line 214 explicitly introduces the third type of coordination. Make it clear which is pattern 2 is and which example(s) illustrate it. Then line 216 “Finally, a very common practice…” so this is a fourth pattern of coordination? I would suggest in Line 169 you state “There were 4 patterns of coordination that varied relative to when the verb occurred relative to the action and whether or not the infant was involved in the action” (or something like this)

Line 236 “Another observation was that the…”-again, try to introduce how many patterns there were and how they varied before describing all of them and giving examples? (ditto for line 247)

Line 257: I like the summary of the observations, but I think it would be helpful for the reader to have some sense before reading all of the patterns and examples how they vary-then the examples are more clear in the reader’s mind as they read through them (in terms of how they vary relative to the study question). (similar to what is done in the Discussion, lines 400-406)

Thank you very much for this suggestion. We have added a paragraph guiding the reader through the qualitative analysis.

Lines 281-283: Instructions to the author seem to be mistakenly included in the manuscript: “This section may be divided by subheadings. It should provide a concise and precise description of the experimental results, their interpretation as well as the experimental conclusions that can be drawn.”

We are sorry for having overlooked this. It has been deleted in the revised version

Line 292: delete “categories” that first occurs in the sentence

Line 293: 69% doesn’t seem to match the graph? I think that Figure 2 is just Figure 3 with different x-axis labels. Please provide the correct Figure 2.

Done

Line 301: change “out” to “up”

Figure 3b: it is interesting that the same pattern of temporal coordination offset by more than two seconds follows temporal coordination within a two second time window. It would be interesting to know which verbs these were (and verb outcomes tested at a later timepoint) and how infants’ gaze (or other behavior) relates to which verbs are synchronous vs. offset (Fig 4 looks at gaze relative to synchrony vs. offset presentation, but specific verbs could be explored)

While we value this point, we think that for this paper, a more fine-grained analysis would take away from our main focus, which is to explore patterns in action-verb coordination. We will follow up the reviewer’s suggestion in our future work.

Line 307: “multimodal practices” again, change to ‘responses’ or ‘behaviors’ I think?

(if you prefer to keep the word practices, it is spelled with a ‘c’ here and an ‘s’ earlier in the manuscript)

Done

ANOVA relating to Figure 4-gaze and synchrony. One interesting question for the verbs that are temporally offset would be to look at the timing of gaze relative to the presentation of the verb and /or action that is offset. Maybe if you look at sequential patterns/time series, you might see that a verb (or action) occurs when the infant looks. So, for example, the mother was only talking (verb utterance), but the infant looks, so she performs the action. The infant gaze ‘elicits’ the presentation of both, which in some cases makes it offset? But maybe this is still better than not being presented together at all?

Our analyses of the verbs that are temporally offset revealed that commonly, these were a response to infants’ actions or the verb was an announcement of an action to be performed. The next step would certainly be to analyze whether infant’s gaze can elicit a particular verb presentation. We agree with the reviewer that for this question, a sequential patterns/time series is the appropriate method to approach it. Again, we hope that the reviewer will understand our decision not to overload the paper with too many analyses.

General question: I realize it is a small sample, but could you look at individual differences in mothers who show synchrony vs. decontextualized? (to address the comment, lines 461-462, “…mothers who made more extensive use of verbs had children who at 24 months…”)

We tried multiple analyses in that direction. We have included a descriptive part in the results with some information on the types of verbs we encountered. According to this typology we tried to somehow relate the mothers' use of verbs with the types of synchrony.

However, we have to admit that we could not see any clear pattern. It could well be that the sample is rather "thin" to allow for any patterns to emerge... 

We think that a more fine-grained analysis would require a different typology of verbs. In addition, from the literature we gained the impression that the concept of “decontextualized” linguistic items is still in progress. In another project, we have begun to define it in a more precise way that accounts for early interactions. For our paper, we think that it is important to identify this type of verbs as being present in the input, because other authors claimed that decontextualized speech start to occur when the children get older.

Also, could you assess the verbs that mothers used vs. the ones that children know at 24 months? Even if not specifically learning the exact verbs mothers used, did infants learn more verbs that lend themselves to synchronous action/verb presentation (vs. more abstract verbs or ones that are less tightly synced to behavior)?

Following this comment, we attempted an analysis in this direction. More specifically we divided the Verbs in the ELFRA parental report into verbs affording synchrony, verbs for complex actions and decontextualized verbs and looked at whether we could find any relationships between mothers’ behavior and infants’ verb production. The regression models replicated the analyses we already report for all the verbs taken together. It didn’t add any new insights. Also due to the fact that the ELFRA includes mostly verbs which afford synchronous presentation we do not have the data to proceed with this analysis. 

We are sorry not to have been able to provide any interesting results for these very very very constructive and interesting comments. They have given us food for thought on how to further continue with the analyses of this data.